# Effectiveness of Computer-Mediated Educational Counseling for Tinnitus Relief: A Randomized Controlled Trial

**DOI:** 10.3390/brainsci14070629

**Published:** 2024-06-24

**Authors:** Sumin Lee, Tae-Jun Jin, Donghyeok Lee, In-Ki Jin

**Affiliations:** 1Department of Speech Pathology and Audiology, Graduate School, Hallym University, Chuncheon 24252, Republic of Korea; eaeno123@gmail.com (S.L.); jtj73872711@gmail.com (T.-J.J.); deggw0918@gmail.com (D.L.); 2Division of Speech Pathology and Audiology, Research Institute of Audiology and Speech Pathology, College of Natural Sciences, Hallym University, Chuncheon 24252, Republic of Korea

**Keywords:** teleaudiology, tinnitus, counseling, rehabilitation

## Abstract

Counseling can help alleviate tinnitus-caused emotional distress and correct misconceptions, making it an effective rehabilitation option for people with tinnitus. Advances in communication technology have increased the demand for computer-mediated tinnitus counseling; however, the effectiveness of such counseling in reducing tinnitus is unclear. Thus, this study aimed to determine the tinnitus-relieving effects of computer-mediated counseling. Thirty-six participants with tinnitus were randomly assigned to online counseling (15 participants) or video-based counseling (21 participants) groups, defining how remote counseling was conducted. Tinnitus counseling, comprising 100 items, lasted 2 weeks and was separated into six sessions for the online counseling group and 8–9 items daily for 12 days for the video-based counseling group. The effectiveness of counseling was determined based on score changes between baseline and 2-week follow-up using the Korean version of the Tinnitus Primary Function Questionnaire and Visual Analog Scales for annoyance and loudness. While no significant improvements were observed in other domains, average emotional aspect-related scores showed significant improvements in both groups. Regarding individual results, four and seven participants in the online and video-based counseling groups reported significant improvements in the emotional domain, respectively. Overall, computer-mediated educational counseling might be a rehabilitation option for individuals with tinnitus.

## 1. Introduction

Tinnitus is the perception of sound without a corresponding external acoustic signal [1]. People of all ages, including the pediatric population, may experience tinnitus, especially during upper respiratory infections, such as severe acute respiratory syndrome coronavirus [2]. The prevalence of tinnitus varies by country, age, and sex, but the general prevalence of tinnitus is 10–15% [3]. Tinnitus can be caused by various causes, including hearing loss, aging, and genetics [4]. Tinnitus can negatively affect the lives of those affected. In a study of 72 people by Tyler and Baker [5], 93.0% reported negative effects on lifestyle, such as sleeping problems, and 69.4% reported difficulties with emotional regulation, including depression, irritation, and confusion. Thus, tinnitus can affect the quality of life through various symptoms.

Although a complete cure for tinnitus is currently not available [6], several treatment options have been proposed to relieve symptoms [7,8,9]. One of these options is counseling [10]. Counseling is a rehabilitation approach that can help alleviate negative emotions, such as irritation, caused by tinnitus and may even help promote tinnitus habituation, a phenomenon in which a person is aware of their tinnitus but is no longer distressed by it [11,12,13].

In the audiology field, two types of counseling are commonly offered. One is educational counseling, which provides patients and families with information about ear-related diseases and tinnitus, explains the effectiveness of various interventions, and teaches coping skills. The other is personal adjustment counseling, which focuses on the feelings, emotions, thoughts, and beliefs expressed by patients, their families, and caregivers and promotes psychological recovery through the counselor’s empathy and understanding of the patient [14]. Of these, educational counseling is the most commonly used approach in tinnitus rehabilitation [4]. The content of educational counseling utilized in tinnitus rehabilitation includes information about tinnitus in general, the physiological and psychological causes of tinnitus, and strategies for lifestyle modification to cope with tinnitus [15,16]. Therefore, educational counseling can help participants understand their symptoms, correct misconceptions about tinnitus, and develop coping strategies [17,18].

Several studies have reported that educational counseling results in psychological relief from tinnitus. For example, Henry et al. conducted four group educational counseling sessions with 94 participants over 4 weeks, each session lasting 90 min, of which 15 min were spent on discussions among the participants [19]. The Tinnitus Severity Index was used to assess tinnitus severity before and 6 months after intervention. The results showed a significant decrease in this score from 24.9 before the intervention to 21.6 after 6 months, indicating an improvement in tinnitus severity. Kröner-Herwig et al. conducted an educational counseling program that consisted of two group sessions with 16 participants with tinnitus [20]. In the first session, education about tinnitus and self-help strategies for coping with tinnitus were provided, and the participants practiced the self-help strategies for 4 weeks, after which the self-help strategies were discussed in the second session. Subjective questionnaires of general well-being (physical well-being, activity, mood, and stress) and tinnitus-related variables (such as loudness, disability, awareness, and control) were used to assess the degree of improvement. The results showed that the educational counseling group improved in all tinnitus-related variables, except for tinnitus loudness, and showed improvements in the “mood” and “coping with stress” domains of overall well-being compared with the no-intervention group.

During the 2020 coronavirus (COVID-19) pandemic, the use of contactless digital technologies increased in many fields compared to pre-pandemic levels [21,22]. Contactless digital technologies have also been used in counseling, with 84.3% of counselors reporting their use after the pandemic, compared with 41.7% before the pandemic. During the pandemic, 84.3% of counselors reported providing counseling via video conferencing and phone calls [23]. Psychologists were expected to perform 34.9% of their clinical work utilizing remote technologies even after the pandemic had ended [24]. Therefore, counseling using contactless digital technology is expected to continue to be used in the future.

Some studies have reported that tinnitus can be alleviated through remote counseling. In a study by Henry et al., 36 participants with tinnitus participated in six educational counseling sessions over 6 months that were conducted via telephone with an audiologist and a psychologist [25]. Comparing the mean of the participants’ pre- and post-counseling scores on the Tinnitus Handicap Inventory (THI), the authors reported a significant decrease from 60.7 at baseline to 47.5 after 6 months, showing the potential of remote counseling for improving tinnitus. The THI is a 25-item self-report questionnaire developed to assess the degree of tinnitus-related handicap on a 100-point scale [26]. For example, a score of 76–58 is rated as a “severe handicap”, whereas a score of 56–38 is rated as a “moderate handicap.” Similarly, Malouff et al. found that self-help books can facilitate reducing distress caused by tinnitus [27]. In their study, 162 participants received a tinnitus self-help book and were instructed to read it and follow the suggestions in the book for 6 weeks without further contact between the researchers and participants until the final assessment. The results showed that the self-help book helped participants experience less distress from tinnitus compared to those who did nothing, with 25% reporting a significant reduction in tinnitus-related distress.

In addition, videos can be used to provide educational counseling. Recently, it has been reported that 74% of adult internet users obtain information about various health issues, including tinnitus, through online [28] and video-sharing services such as YouTube (Google Inc., Mountain View, CA, USA), which is one of the most popular online platforms [29]. Various videos that provide self-help strategies for patients with tinnitus are available on this platform, including self-management videos that distract from tinnitus with sounds, such as white noise and running water, and information-sharing videos that provide information about tinnitus being the most frequently uploaded videos. Among tinnitus-related videos, the three most viewed videos were all related to tinnitus self-management, with 7.1, 5.6, and 2.1 million views, respectively [30]. This indicates that people with tinnitus are willing to utilize video-sharing platforms to self-manage tinnitus. 

Face-to-face educational counseling helps relieve tinnitus symptoms and, in some cases, has been clinically proven to result in psychological relief [11,12,13]. However, many variations exist in how counseling is provided, the content of the counseling, and the way it is conducted. In addition, the effectiveness of remote group educational counseling and computer-delivered educational counseling, such as that provided through videos, in reducing tinnitus remains unclear.

Therefore, this study aimed to determine the effectiveness of computer-mediated educational counseling, which has not been clearly proven to relieve tinnitus. The participants were categorized into two groups: those who received online group counseling using tinnitus-related counseling content and those who watched videos with tinnitus-related educational content uploaded to an online platform. The results of this study may provide useful guidance for hearing professionals, counselors, and people with tinnitus in selecting appropriate and effective counseling methods. 

## 2. Materials and Methods

### 2.1. Participants

The required minimum sample size was calculated using the G*Power 3.1.9.7 statistical software program (Heinrich Heine Universität., Düsseldorf, Rhein-Ruhr, Germany), with parameters set as follows: ES, 0.5; type 1 error, 0.05; and type 2 error, 0.20. The calculated sample size was 27. The sample size was determined to be a minimum of 15 participants per group based on previous studies that reported the effectiveness of tinnitus counseling with 12–16 participants per group [20,25]. All participants were recruited through a tinnitus community website “https://cafe.naver.com/onquest (accessed on 29 June 2023” and screened using the inclusion and exclusion criteria listed in Table 1. The participant selection process was conducted through a telephone interview. The outcome measures and study design sections provide detailed information regarding the interview and tinnitus evaluation. This study was approved by the institutional review board of Hallym University (HIRB-2023-043). Each participant received a written explanation of this study’s aims, protocol, and procedures before providing written informed consent prior to study participation. The consent form included a notice that the participants could stop the study at any time as desired. This clinical trial was registered with the ISRCTN Registry (https://www.isrctn.com, trial number: ISRCTN15292645) on 3 July 2023.

The study participants were recruited from 6 July 2023 to 28 July 2023, and the follow-up examinations were performed between 22 August 2023 and 3 September 2023. The Consolidated Standards of Reporting Trials (CONSORT) flow diagram is shown in Figure 1. Out of 69 applicants, 25 were excluded during the telephone interview process and 8 during the first visit for the following reasons: their average Korean version of the Tinnitus Primary Function Questionnaire (K-TPFQ) scores were ≤30 points (n = 6), they did not respond to the contact (n = 26), or they did not want to proceed with tinnitus counseling for 2 weeks (n = 1). Thus, 36 participants were finally enrolled in this study without further losses or exclusions.

### 2.2. Study Design

The present study was conducted for 2 weeks to determine the effectiveness of remote counseling in alleviating tinnitus through two different methods—online group counseling vs. video-based counseling. Therefore, no other tinnitus treatment or rehabilitation beyond counseling was provided during this study. The study protocol is shown in Figure 2. Two weeks before baseline, a telephone interview was conducted with applicants wishing to participate to confirm the initial screening criteria. The researcher who conducted the telephone interviews was a master’s level researcher with a license as an audiologist. Responses to the K-TPFQ and Visual Analog Scale (VAS) were also obtained during this telephone interview. The researcher explained how to respond to each questionnaire and then read the questions, and the applicant provided the scores. After completing the questionnaires, participants who met the inclusion criteria were scheduled for their initial visit.

During the initial visit, which lasted approximately 1 h, a researcher explained the study, and each participant signed a written informed consent form. Subsequently, each participant completed the tinnitus questionnaires (i.e., Tinnitus Intake Questionnaire [TIQ], K-TPFQ, and VAS). The TIQ is a questionnaire designed to collect basic demographics (e.g., sex and age) and tinnitus-related background information (such as duration of tinnitus and diagnosis result). Participants with a K-TPFQ score of >30 points received a pure-tone audiogram and tinnitogram (loudness and pitch matching). A loudness-matching test compares the loudness of an individual’s perceived tinnitus to an external sound, with the goal of finding the equivalent loudness. The pitch-matching test compares the pitch of an individual’s perceived tinnitus to external sounds to find a similar frequency area. The tests were performed using a GSI AudioStar Pro audiometer (Grason-Stadler Inc., Eden Prairie, MN, USA) with Sennheiser HDA-200 headphones (Sennheiser electronic GmbH & Co. KG, Wedemark, Germany). Pure-tone audiometry was performed to determine the average difference in pure-tone thresholds at 500, 1000, and 2000 Hz between the two groups, and the tinnitogram was performed to confirm that participants had tinnitus and to characterize their tinnitus. Both the pure-tone audiometry and tinnitogram were performed at the initial visit only. The TIQ, VAS, and K-TPFQ were completed in a consultation room, while the pure-tone audiometry and tinnitogram were performed in a double-walled sound booth.

The 36 participants enrolled in this study were randomly assigned to one of the two study groups using the randomization function in Microsoft Excel (Microsoft Corporation, Redmond, WA, USA). The randomization program assigned numbers 1–36 to the 36 participants and ordered them randomly: participants assigned numbers 1–18 were assigned to the online counseling group, and participants assigned numbers 19–36 were assigned to the video-based counseling group. However, three participants in the online counseling group requested to be transferred to the other group owing to personal reasons. Thus, the video-based counseling group comprised 21 participants, whereas the online counseling group included 15 participants. 

After the initial visit, participants received group-specific tinnitus counseling for 2 weeks. The online counseling group was split into subgroups of five participants who were asked to attend counseling via an online conferencing system (Zoom Technologies, Inc., San Jose, CA, USA) 3 days a week (Monday, Wednesday, Friday or Tuesday, Thursday, Saturday) for a total of 6 sessions over 2 weeks. The number of sessions was determined based on a previous study that reported improvements in tinnitus after six sessions of in-person counseling [25]. The counseling session took place at 8 P.M. and was conducted by a certified audiologist who had conducted tinnitus counseling more than 20 times. Each session consisted of the audiologist answering one question and then asking the participants if they had any additional questions or concerns, with the audiologist answering these additional questions.

For the video-based counseling group, participants were asked to watch eight to nine 1–2-min educational videos on tinnitus per day, 6 days a week (Monday through Saturday), for 12 sessions over 2 weeks. The researcher sent the participants a video link and a viewing confirmation survey via a mobile phone message every morning, and participants could watch the video at any time during the day. Participants were considered to have watched a video if they completed the viewing confirmation survey for that video. The viewing confirmation survey was designed to ensure participants had watched the content, making it difficult to complete the survey without having watched the video.

At the last counseling session for both groups, the participants were also asked whether they were satisfied with how the session was conducted and organized, and whether it helped them learn about tinnitus. The satisfaction survey questions were as follows: “How satisfied are you with your counseling session? (counseling satisfaction)” and “Did the counseling help you learn about tinnitus? (education effectiveness).” Each question was answered on a 5-point scale from very dissatisfied or unhelpful (1) to very satisfied or helpful (5).

After 2 weeks of group tinnitus counseling, the researcher conducted a telephone interview with all participants from both groups. The K-TPFQ and VAS scores were measured during the telephone interview. 

### 2.3. Tinnitus Counseling Contents

The tinnitus counseling contents used in the present study were derived from a previous study [31]. These authors collected information from people with tinnitus by asking them questions such as “What do you know about tinnitus, and what would you like to know more about?”, “What treatments do you think are effective for tinnitus?”, and “What difficulties and inconveniences do you experience due to tinnitus?”. Based on the replies, they developed 100 question-and-answer counseling prompts based on literature research and expert opinions. The answers to the questions were validated by three hearing professionals (two professors who majored in audiology and one otolaryngologist). The 100 selected items were categorized into five subcategories: causes (n = 48), symptoms (n = 9), diagnosis (n = 3), prevalence (n = 2), and rehabilitation and treatment of tinnitus (n = 38).

For the materials used in the online group counseling, the 100 counseling items were divided into six PowerPoint materials (Microsoft Corporation, Redmond, WA, USA). Each counseling session consisted of 16–17 questions and answers, and the answers were accompanied by at least one visual to enhance the participants’ memory [19]. For the video-based counseling group, 100 educational counseling videos were created using Adobe Character Animator (Adobe System Inc., San Jose, CA, USA). The videos are organized such that they first show the question to be addressed in the video, and then the counselor character narrates the answer to the question. At least one or two visual examples were included in the videos to enhance the participant’s memory of the answer [19]. At the end of the video, a one-page summary of the answer and a description of the solution that the participant could apply to the question were presented. 

### 2.4. Outcome Measures

The effectiveness of the counseling was assessed based on score changes in the K-TPFQ and VAS scores for annoyance and loudness between baseline and follow-up (2 weeks after baseline) in each group. The K-TPFQ is a tinnitus questionnaire that assesses the negative impact of tinnitus on the quality of life. It is categorized into four subcategories: emotion, hearing, concentration, and sleep, which are typical factors that can be severely affected by tinnitus [32,33]. The K-TPFQ comprises 20 questions, with 5 questions for each subcategory, and each question can be rated on a scale from 0 to 100. A score closer to 0 indicates strong disagreement with the question, while a score closer to 100 indicates strong agreement. The scores for all questions can be averaged to measure overall difficulty with tinnitus, and the average scores can be calculated by subcategory to determine the subcategories causing difficulties. The K-TPFQ can be interpreted as showing clinical tinnitus improvement when the score after rehabilitation decreases by at least 13 points compared with that before [32]. 

The VAS is a scale that asks participants to rate their perceived tinnitus annoyance and loudness on a scale from 0 to 100. The scale consists of a straight line with 0 on the far left and 100 on the far right. Lower scores indicate less annoyance or smaller tinnitus loudness, whereas higher scores indicate more annoyance or larger tinnitus loudness. The VAS score can be interpreted as a clinical tinnitus improvement when the difference between pre- and post-treatment scores is at least 15 points [34].

### 2.5. Data Analysis

To investigate the effectiveness of online group and video-based counseling in relieving tinnitus, the K-TPFQ and VAS scores were recorded at baseline and follow-up. Because it can be difficult to collect reliable data if tinnitus improves or worsens independently from counseling, the K-TPFQ and VAS scores were also measured through a telephone interview 2 weeks before baseline, and the questionnaire scores were compared with those measured at baseline.

The collected data were analyzed using IBM SPSS Statistics for Windows version 26.0 (IBM Corp., Armonk, NY, USA). The paired *t*-test was used in each group to determine significant differences in questionnaire scores at baseline compared with those at 2 weeks before baseline and those at follow-up time points. If the normality of the data was not satisfied, the Wilcoxon signed-rank test was performed. Statistical significance was set at *p* < 0.05. To confirm the clinical effectiveness in individual participants, the change in questionnaire scores of each participant and the minimum clinically important difference (MCID) for each evaluation tool were compared.

## 3. Results

The CONSORT 2010 checklist is provided in Appendix A. Demographic information, tinnitus characteristics, and the results of hearing and tinnitus evaluations of the two groups at baseline are provided in Table 2. The online counseling group included 15 participants (10 women and 5 men) with a mean age of 41.13 years. The average hearing thresholds for the left and right ears were 14.67 and 11.89 dB hearing levels, respectively. The average tinnitus frequency was 4966.67 Hz, the mean tinnitus loudness had a 10.95 dB sensation level, and the mean tinnitus duration was 64.87 months in this group. The video-based counseling group included 21 participants (7 women and 14 men) with an average age of 50.00 years. The average hearing thresholds in the left and right ears were 14.60 and 14.37 dB hearing levels, respectively. In this group, the average tinnitus frequency was 6690.48 Hz, the average tinnitus loudness had a 4.75 dB sensation level, and the average tinnitus duration was 41.43 months. The two study groups did not significantly differ in age, pure-tone average, tinnitus frequency, loudness, or duration. Moreover, no significant between-group differences in the K-TPFQ scores and VAS scores for annoyance and loudness were found at baseline. No participant reported any adverse effect during the study, including increased discomfort owing to tinnitus or hearing loss.

Figure 3 and Figure 4 show the respective average K-TPFQ and VAS scores at 2 weeks before and after baseline for each group. Comparing K-TPFQ and VAS scores between 2 weeks before baseline and at baseline, no significant changes in mean scores were observed in either group. Regarding the emotion subcategory of the K-TPFQ at 2 weeks before baseline and at baseline, the online counseling group and video-based counseling group showed a non-significant score change from 72.33 to 73.53 (*t* = −0.632, *p* = 0.537) and from 72.43 to 71.71 (*t* = 1.019, *p* = 0.320), respectively. In the hearing subcategory of the K-TPFQ, the average scores in the online counseling group and video-based counseling group exhibited a non-significant change from 41.23 to 40.89 (*t* = 0.213, *p* = 0.834) and from 48.00 to 48.76 (*t* = −0.631, *p* = 0.535), respectively. In the concentration subcategory of the K-TPFQ, the score changes were not significant in the online counseling (from 54.83 to 57.20; *t* = −1.952, *p* = 0.071) and video-based counseling (from 56.00 to 54.90; *t* = 0.740, *p* = 0.468) groups. Likewise, the average K-TPFQ scores in the sleep subcategory did not significantly differ in the online counseling group (from 50.80 to 48.07; *t* = 1.940, *p* = 0.073) and video-based counseling group (from 60.10 to 62.00; *t* = −1.124, *p* = 0.274). Consequently, the average total K-TPFQ score did not change significantly in the online counseling (from 54.80 to 54.92; *t* = −0.153, *p* = 0.881) and video-based counseling (from 59.13 to 59.35; *t* = −0.347, *p* = 0.733) groups.

Regarding the average VAS scores for annoyance at 2 weeks before baseline and at baseline, the online counseling and video-based counseling groups showed non-significant changes from 69.67 to 72.67 (*t* = −0.987, *p* = 0.340) and from 72.62 to 74.52 (*p* = 0.275), respectively. Similarly, the average VAS scores for loudness did not significantly change in the online counseling group (from 68.33 to 68.33, *p* > 0.999) and video-based counseling group (from 65.71 to 67.86, *p* = 0.275). Additionally, none of the participants showed a change above the MCID value for the K-TPFQ and VAS.

Figure 3 and Figure 4 also show the average K-TPFQ and VAS scores, respectively, at baseline and follow-up for each group. Comparing the average scores between baseline and follow-up on the emotion subcategory of the K-TPFQ, both online counseling and video-based counseling groups showed significant decreases from 73.53 to 65.07 (*p* = 0.001) and from 71.71 to 61.26 (*p* < 0.001), respectively. In the hearing subcategory of the K-TPFQ, the average scores of the online counseling and video-based counseling groups changed from 40.89 to 41.49 and from 48.76 to 45.71, respectively, however, these changes were not significant (*p* = 0.944 and *p* = 0.050, respectively). In the concentration subcategory of the K-TPFQ, the online counseling and video-based counseling groups reported non-significant changes from 57.20 to 55.07 (*t* = 1.484, *p* = 0.160) and from 54.90 to 51.95 (*p* = 0.079), respectively. For the sleep subcategory of the K-TPFQ, the change from 48.07 to 46.04 in the online counseling group was not significant (*t* = 1.144, *p* = 0.272), whereas the average score changed significantly from 62.00 to 56.71 in the video-based counseling group (*p* = 0.001). The total K-TPFQ score exhibited a non-significant change from 54.92 to 53.17 in the online counseling group (*t* = 1.283, *p* = 0.220), whereas the video-based counseling group had a significant change from 59.35 to 55.16 (*p* = 0.001). 

Comparing baseline with follow-up VAS scores for annoyance, the online counseling and video-based counseling groups showed significant decreases from 72.67 to 68.13 (*t* = 2.220, *p* = 0.043) and from 74.52 to 68.10 (*p* = 0.017), respectively. The average VAS scores for loudness changed from 68.33 to 65.80 in the online counseling group and from 67.86 to 66.43 in the video-based counseling group; but neither difference was significant (*t* = 1.514, *p* = 0.152 and *p* = 0.443, respectively). 

The MCID values for each assessment tool were used to analyze the clinical significance of the tinnitus relief in each participant. The MCID values are 13 points for the K-TPFQ and 15 points for the VAS for annoyance and loudness. In the emotion category of the K-TPFQ, 4 out of 15 (26.67%) participants in the online counseling group and 7 out of 21 (33.33%) participants in the video-based counseling group showed an improvement of 13 points or more. In the hearing category of the K-TPFQ, 0 out of 15 (0%) and 3 out of 21 (14.29%) participants, respectively, showed improvements of at least 13 points. In the concentration category, 1 out of 15 (6.67%) participants in the online counseling group and 3 out of 21 (14.29%) participants in the video-based counseling group improved by 13 points or more. In the sleep category of the K-TPFQ, 1 out of 15 (6.67%) participants in the online counseling group and 3 out of 21 (14.29%) participants in the video-based counseling group showed an improvement of at least 13 points. Regarding the total K-TPFQ score, 0 out of 15 (0%) participants in the online counseling group and 4 out of 21 (19.05%) participants in the video-based counseling group showed an improvement of ≥13 points. Regarding the VAS score for annoyance, 2 out of 15 (13.33%) participants in the online counseling group and 3 out of 21 (14.29%) participants in the video-based counseling group showed an improvement of ≥15 points. Regarding the VAS score for loudness, 1 out of 15 (6.67%) participants in the online counseling group and 2 out of 21 (9.52%) participants in the video-based counseling group showed improvements of ≥15 points.

The survey results of individual participants regarding their satisfaction with the counseling are reported in Appendix A. For the online counseling group, the mean scores for counseling satisfaction and education effectiveness were 4.60 (*SD* = 0.51) and 4.73 (*SD* = 0.46), respectively. For the video-based counseling group, the mean scores were 4.19 (*SD* = 0.75) and 4.57 (*SD* = 0.51), respectively. For counseling satisfaction and effectiveness, the mean scores by group were all above 4, i.e., the participants were generally satisfied or found the counseling helpful. When comparing individual scores, for counseling satisfaction, 32 participants gave a score of 4 or higher (88.89%), except for four participants in the video-based counseling group who assigned a score of 3, and for education effectiveness, all participants assigned a score of 4 or higher (100%). Between-group differences in mean scores were not significant for counseling satisfaction (*t* = 1.834, *p* = 0.075) or education effectiveness (*t* = 0.983, *p* = 0.333).

## 4. Discussion

The main objective of this study was to determine the effectiveness of tinnitus relief through remote educational counseling. Thirty-six participants with tinnitus received either online group-based or video-based counseling, and tinnitus relief was measured based on changes in K-TPFQ and VAS scores. Each questionnaire was administered three times: 2 weeks before baseline, at baseline, and at follow-up (2 weeks after baseline). In both groups, no significant differences in questionnaire scores were found between baseline and 2 weeks before baseline, suggesting that participants experienced tinnitus consistently before counseling. However, both groups showed significant improvements in comparing baseline and follow-up time points in some assessment areas. In the video-based counseling group, significant improvements were found in the emotion and sleep categories of the K-TPFQ and in the VAS scores for annoyance, whereas the online counseling group showed significant improvements in the emotion category of the K-TPFQ and in the VAS scores for annoyance. Although not all study participants showed significant tinnitus relief, the emotional relief common to both groups demonstrates the potential for remote educational counseling to contribute to tinnitus relief in the field of emotional care. 

The tinnitus educational counseling provided in this study generally helped participants decrease tinnitus-related emotional distress but had less effect on changes in tinnitus loudness. For example, 30.6% of all participants showed improvements above the MCID for the emotion subcategory of the K-TPFQ, whereas only 8.3% showed improvements exceeding the MCID for the VAS assessing loudness. These findings are similar to those of previous studies that examined the effectiveness of tinnitus educational counseling. Wang et al. [35] studied 51 participants with tinnitus and found that after three or more tinnitus educational counseling sessions, 24 (47.06%) participants had a decrease in THI score points of ≥ 20, confirming that educational counseling can help improve tinnitus reactions including emotional aspects. A study by Schlee et al. used a smartphone application without professional intervention to provide participants with tinnitus-related educational content for 4 months [36]. These authors reported no change in tinnitus loudness, but small-to-medium improvements in THI effect size, indicating improvement in tinnitus-related distress. The results of the current study and related prior research indicate that tinnitus educational counseling may be more likely to improve reactions to tinnitus, including emotional distress, than to change tinnitus loudness. Therefore, tinnitus educational counseling may be a viable option for patients with tinnitus who need emotional relief.

Currently, clear guidelines for tinnitus educational counseling do not exist [31]. The content of tinnitus educational counseling used in different studies varies widely, and the effectiveness of the counseling varies depending on the educational counseling content. Therefore, this study conducted a survey to determine whether the content of the educational counseling was satisfactory and helpful to the participants (Appendix A). The high scores for counseling satisfaction and education effectiveness in both groups suggest that the educational counseling provided in this study was generally considered satisfactory and effective for the participants.

In the present study, both groups showed improvements mainly in emotional aspects. However, it is important to be cautious about interpreting this as an indication that all patients who receive educational counseling will improve emotionally. It might be important to consider the MCID alongside the mean value to interpret the results of this study correctly. When comparing individual improvement scores, 11 out of 36 (30.56%) participants in the emotion category of the K-TPFQ and 5 out of 36 (13.89%) participants in the VAS for annoyance showed a change in at least the MCID value, which can be interpreted as a clinically significant improvement in emotional aspects. This suggests that educational counseling might help improve emotional aspects, but not for every patient.

The present study has some limitations. First, this study found tinnitus relief in both groups in computer-mediated settings, confirming the potential for remote tinnitus counseling to help reduce the negative emotional impact of tinnitus. However, it did not provide a direct comparison to face-to-face counseling. Therefore, it was not possible to confirm whether the remote counseling in this study showed equivalent or different tinnitus relief than face-to-face counseling. A comparative follow-up study between face-to-face and computer-mediated counseling groups is needed to assess this. The findings of this follow-up study will aid in our understanding of the effectiveness of tinnitus relief provided by computer-mediated counseling compared with that of face-to-face counseling. Second, this study measured the effectiveness of computer-mediated counseling in relieving tinnitus by presenting 100 pieces of identical content to participants. However, future studies should measure the effectiveness of counseling using personalized content. Personalized counseling has the potential to save time and increase participants’ motivation for counseling. Third, this study did not investigate the cause of participants’ tinnitus; however, the effectiveness of this intervention may be influenced by the specific medical conditions that cause tinnitus. Therefore, future studies that categorize groups according to the cause of tinnitus and measure the effectiveness of tinnitus relief provided by counseling may help to overcome the limitations of this study. Finally, the present study only measured tinnitus relief effects at the end of the counseling period and did not determine whether these counseling effects were sustained. One study of tinnitus counseling showed significant tinnitus relief compared to the end of the study despite no other intervention for 12 months after the study had ended [19]. Measuring participants’ tinnitus relief after some time would allow us to determine the long-term effectiveness of remote counseling in further studies.

The results of this study have the following practical implications. There are many subtypes of tinnitus; thus, treatment or rehabilitation approaches should be tailored to the specific subtype for effective improvement [37]. The findings of this study suggest that educational counseling is an effective approach for tinnitus patients who report emotional challenges, and that computer-mediated counseling can provide an alternative method of support, especially for individuals unable to attend face-to-face counseling.

## 5. Conclusions

The present study showed that tinnitus educational counseling, delivered through online group counseling and viewing video content, can contribute to alleviating the negative emotional impact of tinnitus. The results of this study provide a valid basis for hearing professionals and counselors to recommend remote tinnitus counseling, depending on the patient’s situation.

## Figures and Tables

**Figure 1 brainsci-14-00629-f001:**
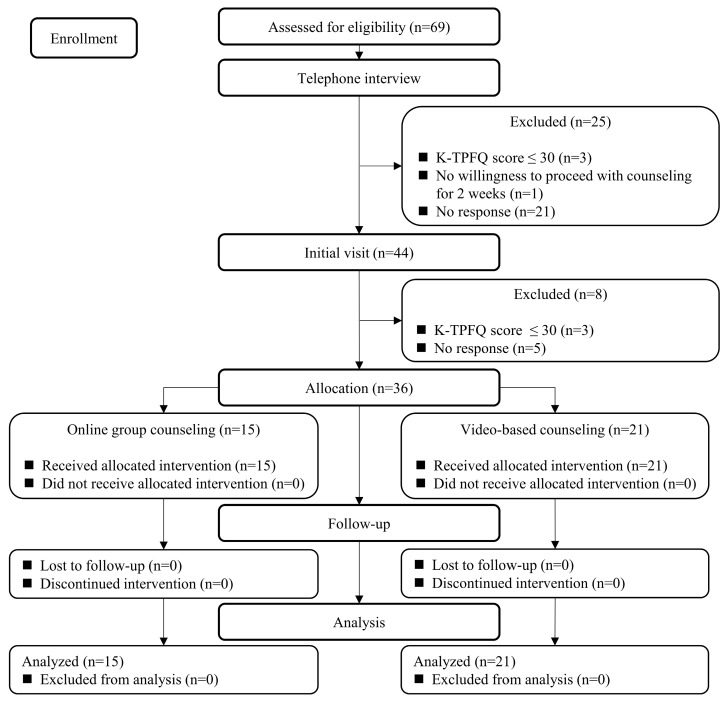
Consolidated Standards of Reporting Trials (CONSORT) flow diagram for this clinical trial.

**Figure 2 brainsci-14-00629-f002:**
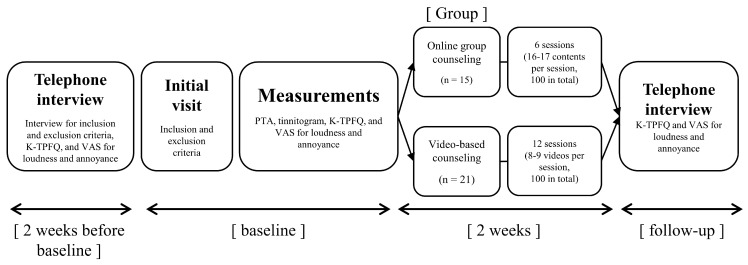
Protocol of the present study. K-TPFQ = Korean version of the Tinnitus Primary Function Questionnaire; PTA = pure-tone audiometry; VAS = Visual Analog Scale.

**Figure 3 brainsci-14-00629-f003:**
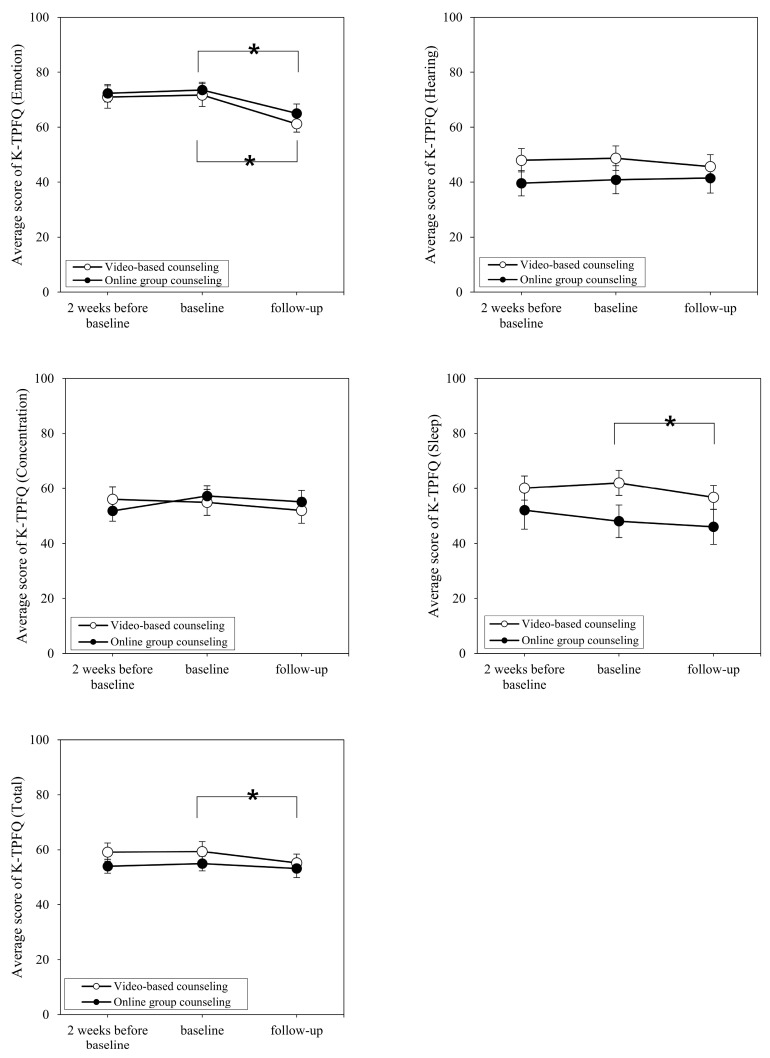
Change in average K-TPFQ scores over time for the two study groups. K-TPFQ = Korean version of the Tinnitus Primary Function Questionnaire. * Asterisks indicate statistical significance.

**Figure 4 brainsci-14-00629-f004:**
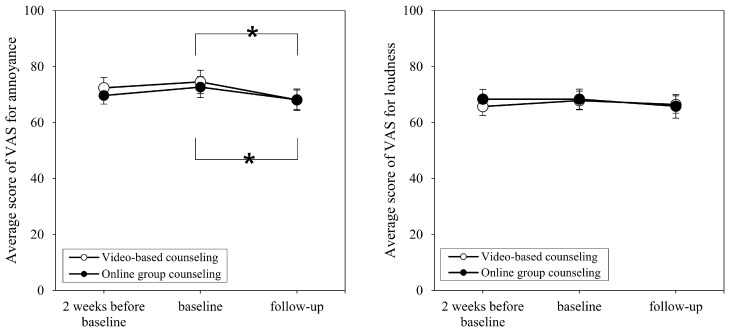
Change in average VAS scores over time for the two study groups. VAS = Visual Analog Scale. * Asterisks indicate statistical significance.

**Table 1 brainsci-14-00629-t001:** Inclusion and exclusion criteria.

Inclusion criteria	Presence of discomfort or difficulty due to tinnitus
	Average score on the Korean version of the Tinnitus Primary Function Questionnaire (K-TPFQ) > 30 points
	Currently not receiving other tinnitus treatments and counseling
	Familiarity with smartphones or the internet
Exclusion criteria	Having psychiatric illnesses
	Involved in tinnitus-related litigation
	Somatosensory tinnitus

**Table 2 brainsci-14-00629-t002:** Participant demographics and tinnitus characteristics, PTA, tinnitus loudness level, K-TPFQ scores, and VAS scores for annoyance and loudness in the two study groups at baseline.

		Group		*p* Value
		Online Group Counseling	Video-Based Counseling	
Sex (number of participants)	Female	10	7	
	Male	5	14	
	Total	15	21	
Age (years)		41.13 (17.09)	50.00 (11.08)	0.092
PTA	Left	14.67 (11.78)	14.60 (8.64)	0.984
	Right	11.89 (12.84)	14.37 (10.49)	0.528
Tinnitus frequency (Hz)		4966.67 (3340.84)	6690.48 (2431.44)	0.087
Tinnitus loudness level (dB SL)		10.95 (12.60)	4.75 (5.31)	0.051
Tinnitus duration (months)		64.87 (120.95)	41.43 (36.21)	0.406
K-TPFQ score	Emotion	73.53 (10.97)	71.71 (18.94)	0.719
	Hearing	40.89 (19.82)	48.76 (20.29)	0.255
	Concentration	57.20 (14.63)	54.90 (21.58)	0.723
	Sleep	48.07 (22.73)	62.00 (21.08)	0.067
	Total	54.92 (10.21)	59.35 (16.64)	0.331
VAS score for annoyance		72.67 (14.50)	74.52 (18.97)	0.752
VAS score for loudness		68.33 (13.97)	67.86 (15.38)	0.925

Note. Data are presented as means (standard deviations). K-TPFQ = Korean version of the Tinnitus Primary Function Questionnaire; PTA = pure-tone average; SL = sensation level; VAS = Visual Analog Scale.

## Data Availability

The data presented in this study are available on request from the corresponding author. The data are not publicly available due to their containing information that could compromise the privacy of research participants).

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
