# Peer review of "Effectiveness of Computer-Mediated Educational Counseling for Tinnitus Relief: A Randomized Controlled Trial"

_brainsci, 2024, doi:10.3390/brainsci14070629_

Round 1
Reviewer 1 Report
Comments and Suggestions for Authors
I would like to congratulate the authors for their accuracy and consistency in conducting the research under review.
Considering thirty-six patients with functionally impacting tinnitus, the counselling for tinnitus relief, either through videos or through online meetings with experienced audiologist, improves the reaction to tinnitus by reducing emotional distress in the short term. In contrast, the effect on tinnitus loudness was weaker.
The manuscript is worthy of publication, nevertheless, major revisions are mandatory since it is unacceptably long (19 pages), extremely detailed and with unnecessary graphic depictions. Hence, some parts should be revised or removed
Author Response
Reviewer 1 comments
I would like to congratulate the authors for their accuracy and consistency in conducting the research under review.
Considering thirty-six patients with functionally impacting tinnitus, the counselling for tinnitus relief, either through videos or through online meetings with experienced audiologist, improves the reaction to tinnitus by reducing emotional distress in the short term. In contrast, the effect on tinnitus loudness was weaker.
The manuscript is worthy of publication, nevertheless, major revisions are mandatory since it is unacceptably long (19 pages), extremely detailed and with unnecessary graphic depictions. Hence, some parts should be revised or removed
Response: Thank you for the careful review of our manuscript and insightful comments. We agree with your comment and have accordingly removed Figures 3, 4, 7, and 8. We removed these four figures but have retained the text describing the main points made by these figures. We have also removed descriptions that were too detailed and could have been omitted (e.g., if a participant did not complete the survey by 6 p.m., the researcher sent an additional text message to encourage them to watch the video.). We have reduced the page count by 5 pages, from pages 19 to 14, even though we included additional content as requested by the other two reviewers. We hope our efforts are in line with your expectations. The numbering of references and figures has been revised by double-checking the references deleted and added on the basis of reviewers' comments.
Reviewer 2 Report
Comments and Suggestions for Authors
I would like to thank the authors for their submission and allowing me to review their work.
This is an interesting study on an important topic. However, I would be grateful if you could add further explanations and changes on the following points:
1) ABSTRACT: Page 1, line 14
Please specify the mean age (± standard deviation) and gender of the study population.
2) ABSTRACT: Page 1, line 16
When was the study conducted?
3) INTRODUCTION: Page 2, line 6
In the introduction section, I suggest spending a few words to explain the causes of tinnitus (I suggest citing the following article: “Tinnitus Update. J Clin Neurol. 2021;17(1):1-10. doi:10.3988/jcn.2021.17.1.1”) and specify that all ages, including the pediatric population (especially during upper respiratory infection, such as SARS-COV-2) may complain of tinnitus (I suggest citing the following article: “Audiological and vestibular symptoms following SARS-CoV-2 infection and COVID-19 vaccination in children aged 5-11 years. Am J Otolaryngol. 2023;44(1):103669. doi:10.1016/j.amjoto.2022.103669”)
4) INTRODUCTION: Page 2, line 84
I suggest explaining the main characteristics of Tinnitus Handicap Inventory (THI).
5) MATERIALS AND METHODS: Page 3, line 126
Were patients with somatosensory tinnitus included or excluded from the study?
6) MATERIALS AND METHODS: Page 3, line 126
Were patients with conductive hearing loss included or excluded from the study?
7) MATERIALS AND METHODS: Page 5, line 175
I suggest better explaining what the loudness and pitch of a tinnitus are.
8) MATERIALS AND METHODS: Page 5, line 175
Were distortion product otoacoustic emissions (DPOAE) and tympanometry performed?
9) DISCUSSION: Page 17, line 501
What are the future prospects of this study?
10) DISCUSSION: Page 17, line 501
What are the practical implications of this study?
Comments on the Quality of English Language
Minor editing of English language is required
Author Response
Reviewer 2 comments
I would like to thank the authors for their submission and allowing me to review their work. This is an interesting study on an important topic. However, I would be grateful if you could add further explanations and changes on the following points:
1) ABSTRACT: Page 1, line 14
Please specify the mean age (± standard deviation) and gender of the study population.
Response: Owing to the word limit of the abstract (200 words), we are unable to add the information mentioned. Currently, the word count of the abstract is 198. Nonetheless, we have incorporated your valuable comments in the main text, and the information you mentioned is described in Table 2.
2) ABSTRACT: Page 1, line 16
When was the study conducted?
Response: Thank you for your valuable comment. Owing to the word limit of the abstract (200 words), we are unable to add the information mentioned. The current word count of the abstract is 198 words, but the information mentioned is included in the main text, reflecting your comment.
Described in the main text: The study participants were recruited from July 6, 2023, to July 28, 2023, and the follow-up examinations were performed between August 22, 2023, and September 3, 2023
3) INTRODUCTION: Page 2, line 6
In the introduction section, I suggest spending a few words to explain the causes of tinnitus (I suggest citing the following article: “Tinnitus Update. J Clin Neurol. 2021;17(1):1-10. doi:10.3988/jcn.2021.17.1.1”) and specify that all ages, including the pediatric population (especially during upper respiratory infection, such as SARS-COV-2) may complain of tinnitus (I suggest citing the following article: “Audiological and vestibular symptoms following SARS-CoV-2 infection and COVID-19 vaccination in children aged 5-11 years. Am J Otolaryngol. 2023;44(1):103669. doi:10.1016/j.amjoto.2022.103669”)
Response: We have addressed your comments and added references accordingly.
Revised: Tinnitus is the perception of sound without a corresponding external acoustic signal [1]. People of all ages, including the pediatric population, may experience tinnitus, especially during upper respiratory infections, such as severe acute respiratory syndrome coronavirus 2 [2]. The prevalence of tinnitus varies by country, age, and sex, but the general prevalence of tinnitus is 10–15% [3]. Tinnitus can be caused by various causes, including hearing loss, aging, and genetics [4].
[2] Aldè, M.; Di Berardino, F.; Ambrosetti, U.; Barozzi, S.; Piatti, G.; Zanetti, D.; Pignataro, L.; Cantarella, G. Audiological and Vestibular Symptoms Following SARS-CoV-2 Infection and COVID-19 Vaccination in Children Aged 5–11 Years. Am. J. Otolaryngol. 2023, 44, 103669. DOI: 10.1016/j.amjoto.2022.103669.
[4] Han, B.I.; Lee, H.W.; Ryu, S.; Kim, J.S. Tinnitus Update. J. Clin. Neurol. 2021, 17 (1), 1–10. DOI: 10.3988/jcn.2021.17.1.1.
4) INTRODUCTION: Page 2, line 84
I suggest explaining the main characteristics of Tinnitus Handicap Inventory (THI).
Response: According to your comment, we have added a brief description of THI.
Revised: The THI is a 25-item self-report questionnaire developed to assess the degree of tinnitus-related handicap on a 100-point scale [26]. For example, a score of 76–58 is rated as a “severe handicap,” whereas a score of 56–38 is rated as a “moderate handicap.”
[26] Newman, C.W.; Jacobson, G.P.; Spitzer, J.B. Development of the Tinnitus Handicap Inventory. Arch. Otolaryngol. Head Neck Surg. 1996, 122, 143–148. DOI: 10.1001/archotol.1996.01890140029007.
5) MATERIALS AND METHODS: Page 3, line 126
Were patients with somatosensory tinnitus included or excluded from the study?
Response: We appreciate your comment. The presence of somatosensory tinnitus was an exclusion criterion and has accordingly been added to the list of exclusions in Table 2.
Revised: Exclusion criteria: Having psychiatric illnesses; Involved in tinnitus-related litigation; Somatosensory tinnitus
6) MATERIALS AND METHODS: Page 3, line 126
Were patients with conductive hearing loss included or excluded from the study?
Response: The presence of conductive hearing loss was not considered in this study; therefore, both inclusion and exclusion criteria were deemed non-inclusive. However, we agree with the importance of your comment and have included it in the limitations of our study.
Revised: Third, this study did not investigate the cause of participants' tinnitus; however, the effectiveness of this intervention may be influenced by the specific medical conditions that cause tinnitus. Therefore, future studies that categorize groups according to the cause of tinnitus and measure the effectiveness of tinnitus relief provided by counseling may help to overcome the limitations of this study.
7) MATERIALS AND METHODS: Page 5, line 175
I suggest better explaining what the loudness and pitch of a tinnitus are.
Response: Thank you for your valuable comment. We have added the purpose of the loudness-matching and pitch-matching tests.
Revised: A loudness-matching test compares the loudness of an individual's perceived tinnitus to an external sound, with the goal of finding the equivalent loudness. The pitch-matching test compares the pitch of an individual's perceived tinnitus to external sounds to find a similar frequency area.
8) MATERIALS AND METHODS: Page 5, line 175
Were distortion product otoacoustic emissions (DPOAE) and tympanometry performed?
Response: We did not perform either test because participant characteristics identified by both tests (such as middle ear diseases) were less relevant to the inclusion and exclusion criteria for this study.
9) DISCUSSION: Page 17, line 501
What are the future prospects of this study?
Response: Thank you for your comment. We have added a statement about the associated future prospects in the limitation statement.
Revised: The present study has some limitations. First, the study found tinnitus relief in both groups in computer-mediated settings, confirming the potential for remote tinnitus counseling to help reduce the negative emotional impact of tinnitus. However, it did not provide a direct comparison to face-to-face counseling. Therefore, it was not possible to confirm whether the remote counseling in this study showed equivalent or different tinnitus relief than face-to-face counseling. A comparative follow-up study between face-to-face and computer-mediated counseling groups is needed to assess this. The findings of this follow-up study will aid in our understanding of the effectiveness of tinnitus relief provided by computer-mediated counseling compared with that of face-to-face counseling. Second, this study measured the effectiveness of computer-mediated counseling in relieving tinnitus by presenting 100 pieces of identical content to participants. However, future studies should measure the effectiveness of counseling using personalized content. Personalized counseling has the potential to save time and increase participants’ motivation for counseling.
10) DISCUSSION: Page 17, line 501
What are the practical implications of this study?
Response: We deeply appreciate your comments. Accordingly, we have added a paragraph on practical implications of the findings.
Revised: The results of this study have the following practical implications. There are many subtypes of tinnitus; thus, treatment or rehabilitation approaches should be tailored to the specific subtype for effective improvement [37]. The findings of this study suggest that educational counseling is an effective approach for tinnitus patients who report emotional challenges, and that computer-mediated counseling can provide an alternative method of support, especially for individuals unable to attend face-to-face counseling.
[37] Jin, I.-K.; Tyler, R.S. Measuring Tinnitus in Pharmaceutical Clinical Trials. J. Acoust. Soc. Am. 2022, 152, 3843. DOI: 10.1121/10.0014699.
Reviewer 3 Report
Comments and Suggestions for Authors
The authors studied the effectiveness of computer-mediated educational counselling on tinnitus patients. Online counselling and video-based counselling have been conducted in two groups of patients. The study was interesting and the results showed that computer-mediated educational counselling may be an option in the management of the patient with this condition. The methodology has been described in detail and the results was well shown. The limitations of the study were highlighted.
Other comments:
1. It seems that the cause of tinnitus in the patients was not investigated. The results of the intervention may be affected by certain diseases causing tinnitus.
2. Appendix B is not shown.
Author Response
Reviewer 3 comments
The authors studied the effectiveness of computer-mediated educational counselling on tinnitus patients. Online counselling and video-based counselling have been conducted in two groups of patients. The study was interesting and the results showed that computer-mediated educational counselling may be an option in the management of the patient with this condition. The methodology has been described in detail and the results was well shown. The limitations of the study were highlighted.
Other comments:
- It seems that the cause of tinnitus in the patients was not investigated. The results of the intervention may be affected by certain diseases causing tinnitus.
Response: We agree with your concerns and have incorporated your comments into the limitations of this study.
Revised: Third, this study did not investigate the cause of participants' tinnitus; however, the effectiveness of this intervention may be influenced by the specific medical conditions that cause tinnitus. Therefore, future studies that categorize groups according to the cause of tinnitus and measure the effectiveness of tinnitus relief provided by counseling may help to overcome the limitations of this study.
- Appendix B is not shown.
Response: Thank you very much for your thoughtful comments. We had uploaded two appendix files, but it appears that only one file was uploaded. To address this issue, we have combined the two appendices into one file and uploaded it.
Round 2
Reviewer 1 Report
Comments and Suggestions for Authors
Your efforts were in accordance with our expectations. Worthy of publication